ecology

cattle and sheep, root ingrowth core, root turnover, *Stipa krylovii* Roshev, grazing management, live and dead root

**Authors for correspondence:**
Chengjie Wang
e-mail: nmgcjwang3@163.com
Guodong Han
e-mail: nmghanguodong@163.com

# Impacts of mixed-grazing on root biomass and belowground net primary production in a temperate desert steppe

Zhanyi Wang[1,2,3], Jing Jin[1,4], Yanan Zhang[1], Xiaojuan Liu[1,5], Yongling Jin[1], Chengjie Wang[1] and Guodong Han[1]

[1]College of Grassland, Resource and Environment, Inner Mongolia Agricultural University, Hohhot 010011, Inner Mongolia, People's Republic of China
[2]Key Laboratory of Grassland Resources, Ministry of Education of the P.R. China
[3]Key Laboratory of Forage Cultivation, Processing and High Efficient Utilization, Ministry of Agriculture of the P.R. China
[4]Inner Mongolia Yili Industrial Group Co., Ltd, Hohhot, 010110 Inner Mongolia, People's Republic of China
[5]Wulanchabu Agriculture and Animal Husbandry Bureau, Wulanchabu, 012000 Inner Mongolia, People's Republic of China

ZW, 0000-0002-9690-878X; CW, 0000-0001-7074-9971; GH, 0000-0002-9311-3607

The impacts of large herbivores on plant communities differ depending on the plants and the herbivores. Few studies have explored how herbivores influence root biomass. Root growth of vegetation was studied in the field with four treatments: sheep grazing alone (SG), cattle grazing alone (CG), mixed grazing with cattle and sheep (MG) and no grazing (CK). Live and total root biomasses were measured using the root ingrowth core and the drilling core, respectively. After 2 years of grazing, total root biomass showed a decreasing trend while live root biomass increased with time during the growing seasons. Belowground net primary production (BNPP) among the treatments varied from $166 \pm 32$ to $501 \pm 88$ g m$^{-2}$ and root turnover rates (RTR) varied from $0.25 \pm 0.05$ to $0.70 \pm 0.11$ year$^{-1}$. SG had the greatest BNPP and RTR, while the CG had the smallest BNPP and RTR. BNPP and RTR of the MG treatment were between those of the CG and SG treatments. BNPP and RTR of the CK were similar to MG treatment. Compared with other treatments, CG had a greater impact on dominant tall grasses species in communities. SG could decrease community diversity. MG eliminated the disadvantages of single-species grazing and was beneficial to community diversity and stability.

# 1. Introduction

Plant root and belowground net primary productivity (BNPP) play an important role in the carbon cycle, and through the carbon cycle they ultimately contribute to the effects of climate change on terrestrial ecosystems, especially for grassland ecosystems in arid and semiarid environments, which account for 55% of terrestrial ecosystems. Rangeland accounts for nearly 40% of China's land area and is an important carbon sink [1,2]. Resource allocation between above- and belowground components in grassland ecosystems is the main factor affecting carbon distribution [3]. In a global survey of temperate grasslands, it was found that belowground biomass averaged 1400 g m$^{-2}$ and that 83% of roots occur in the top 30 cm of soil [4]. Root biomass of grasslands has been reported to exceed aboveground standing biomass by factors of between 2 and 30 [3,5–8]. Accurately estimating root mass and BNPP is important for our understanding of belowground processes affecting the carbon cycle [9].

It is more difficult to measure roots than aboveground parts of plants as they are buried in the soil. Therefore, researchers have tried many methods to study the roots, including soil core samples, pit, ingrowth cores, carbon isotopes and root windows, among others. However, each method has its strengths and weaknesses and insight into root dynamics might be obtained by comparing results from more than one sampling method.

Grazing has an influence on root growth, BNPP and soil carbon. Grazing can affect the stock and flow of carbon between above- and belowground vegetation layers [10]. Herbivory impacts on belowground biomass both directly and indirectly by altering plant species composition and affecting aboveground net primary productivity (NPP), the difference between carbon fixed by photosynthesis and carbon lost to autotrophic respiration [11,12]. Based on a review of published data, Milchunas & Lauenroth [13] concluded that grazing has in most cases a stimulatory effect or at least no negative effects on BNPP. Gao *et al.* [14] reported that heavy grazing reduced belowground biomass and BNPP in one of two years of their experiments. Other investigations show that belowground biomass in grazed areas is equal to or greater than in non-grazed areas, depending on the season of the year [10,15]. Based on the above reports, we found that grazing can increase, decrease or have no effect on BNPP. The above research has mostly either focused on the effects of grazing intensity or merely compared grazed with non-grazed areas. These studies typically explore the 'grazing optimization hypothesis' or an 'intermediate disturbance hypothesis'. Few studies have investigated the effect of herbivore species on root growth and BNPP. Mixed grazing by cattle and sheep is a common feature of farm systems in the grassland area. Mixed cattle and sheep grazing can increase the output per unit area and enhance plant diversity, improve animal performance, reduce parasitism and lead to efficient use of resources [16–24]. These advantages may be due to cattle and sheep having different preferences for plants both with respect to species and plant parts selected during grazing, and a reduction in sward rejection due to dung contamination [25].

The desert steppes of Inner Mongolia are representative of desert steppes in the Eurasian grassland belt, and they account for about 11% (8.4 million ha) of Inner Mongolia's grassland area (78.8 million ha). Sheep and cattle are the main livestock in Inner Mongolia and often graze together. Despite the wealth of previous research on aboveground dynamics in these grasslands, there has been little research on belowground components of desert steppe ecosystems, especially studies comparing differences in the impact of grazing by different herbivore species on vegetation. Therefore, the goals of this research were: to characterize the distribution of root biomass during a growing season in an Inner Mongolian steppe grassland, and (2) to evaluate the effects of herbivore assembly on root biomass, BNPP and RTR, using two sampling methods. We expected that moderate grazing would increase root biomass and BNPP but that there would be differences between herbivore assemblies arising from the varied intake behaviours of the two animal species studied.

# 2. Material and methods

## 2.1. Study sites

This study was conducted in Xilamuren desert steppe, (41°21′ N, 111°112′ E) near the city of Baotou in Damao County, Inner Mongolia, China. The elevation of the study site is 1602 m. The climate is temperate continental, with a short growing season (from May to September) and a long cold winter. Annual precipitation is about 284 mm, about 75% of which falls between July and September. Annual

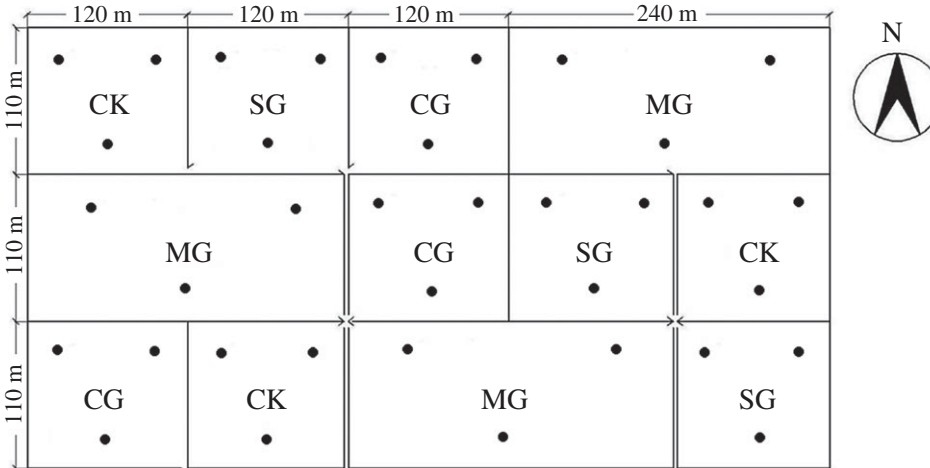

**Figure 1.** The experimental paddock and block locations.

pan evaporation is 2305 mm. Mean annual temperature is 2.5°C. The soil type at the site is Kastannozem (using the FAO classification) with a loamy sand texture. The soil layer is thin (ca 30 cm on average) with stone (calcic horizon) under the soil layer. The dominant species in this steppe vegetation is *Stipa krylovii* Roshev, a perennial grass that is common in the eastern Eurasian steppes and is tolerant to drought and cold. Other common species include the Gramineae *Leymus chinensis* (Trin.) Tzvel, *Cleistogenes songorica* (Roshev.) Ohwi, *Agropyron cristatum* (L.) Gaertn., the forbs *Artemisia frigida* (Willd.), *Heteropappus altaicus* (Willd.) Novopokr., *Potentilla acaulis* L., *Potentilla bifurca* L., *Allium tenuissimum* L., *Allium mongolicum* Turcz. ex Regel, *Kochia prostrata* (L.) Schrad. and *Convolvulus ammannii* Desr. and the legumes *Hedysarum brachypterum* Bunge, *Medicago ruthenica* (L.) Trautv. and *Astragalus galactites* Pall.

## 2.2. Experimental design

The grazing experiment commenced in 2014 and included four treatments: cattle grazing (CG, 3 cattle per plot), sheep grazing (SG, 15 sheep per plot), mixed grazing of cattle and sheep (MG, 3 cattle and 15 sheep per plot) and no grazing (CK). The grazing years were 2014, 2015 and 2016. These treatments were laid out in a randomized complete block design with 3 blocks and 12 fenced paddocks in total (figure 1). Each plot was grazed in the first 7 days of each month from June to September (growing season). During this period, sheep and cattle grazed the respective plots for 24 h per day on seven consecutive days, after which all the animals were moved off the experiment site and grazed elsewhere. Sheep used were 1.5-year-old Mongolian sheep (body weight 45.0 ± 2.0 kg, mean ± s.e.) and cattle were adult Mongolian cattle (body weight 230.0 ± 5.5 kg, mean ± s.e.). Based on livestock intake, species composition and ground cover in the growing season, this stocking rate can be considered to represent moderate grazing intensity with about 25–30% of aboveground plant biomass consumed by the herbivores.

Root biomass was measured in 2016, as described below. No root sampling was conducted in 2015 because the treatment duration was too short. Before the grazing experiment began on these plots in 2014, the area had been under free grazing.

## 2.3. Root sampling

Two methods were used to characterize root dynamics at the same time: drilling core and root ingrowth core. Live root biomass was measured using the root ingrowth core while total root biomass was measured using the drilling core. Root biomass was monitored by drilling holes and collecting and sampling the soil recovered. Root biomass was sequentially sampled during each month of the growing season (from May to September), using an auger of 6 cm in diameter. In each paddock, there were six drilling core sites and three ingrowth core sites, with the sampling sites for drilling cores about 1 m away from the root ingrowth core sampling sites. Samples were separated at collection into three soil layers: 0–10 cm, 10–20 cm and 20–30 cm in depth. For the drilling cores, at each site, two samples made to the same depth were mixed into one sample. For all samples, roots were recovered

from the soil using a 0.2 mm mesh sieve, and running water. All non-root debris (e.g. soil fauna and obvious aboveground stalks, etc.) was removed by hand.

Root ingrowth cores were also used to monitor root biomass and root growth. The ingrowth core method was used according to Milchunas *et al.* [26]. It contains two cylinders of a geometric toroid shape when viewing the concentric cylinders from above the ground. The diameters of outer and inner cylinders were 17 cm and 11 cm, respectively. The cylinders were 30 cm deep. The outside cylinder was made of ridged mesh with 4 mm$^2$ holes. The diameter of most roots was less than 2 mm, based on the measurement of minirhizotron at the site. The first root samples taken in May were thought to be dead root and were not reported. Roots were sampled by removing the sand bags and lifting out the inner cylinders. The mixture of soil and roots between the outer and inner cylinders were dug out and roots were recovered from the soil using a 1 mm sieve in the field. The space between outer and inner cylinders was filled with root-free sifted soil and packed to a similar bulk density as the soil outside the outer cylinder. All root samples were washed in the laboratory using the same methods as described above.

## 2.4. Vegetation survey

Quadrats were sampled diagonally across each paddock to investigate vegetation growth in August 2016. In each fenced paddock, twenty 0.5 m × 0.5 m quadrats for the MG treatment and 16 quadrats for the other treatments were set for monitoring the aboveground biomass. All plants in the quadrats were cut to a stubble height of 1 cm. Parameters measured included vegetation cover, height (estimated using measurements on five to seven plants of each species), shoot biomass and botanical composition of the community.

All the plant samples were weighed after being dried at 70°C for 48 h. According to Huang & Cheng [27], the ash content of the root samples in this area is 12.85%.

## 2.5. Statistical analysis

BNPP (g m$^{-2}$ yr$^{-1}$) was estimated using three methods based on Sims & Singh [3]. The methods were: (I) summation of positive increases in total root biomass (0–30 cm) obtained by the drilling methods; (II) Summation of positive increases in root biomass by depth obtained by the drilling methods and (III) summation of positive increases in live root biomass obtained by the ingrowth core method. Root turnover rates (RTR) were calculated by dividing the annual increment (BNPP) by peak total root biomass [3].

All data analysis was carried out in SPSS software version 24.0 (SPSS Inc., Chicago, IL, USA). Least Significant Difference (LSD) and Tamhane's T2 were used for multiple comparisons of significant differences among treatments. The significance level was set at 0.1 as it was a field trial. When data were not normally distributed and group variances were unequal, then the Kruskal–Wallis Test was used for comparison of means among treatments. The figures were drawn with Excel 2010 and Origin 7.5.

# 3. Results

## 3.1. Live root biomass and total root biomass

The results for root biomass assessed using the ingrowth method are shown in figure 2*a*. Live root biomass in each month on average varied from 49.21 ± 3.07 to 187.64 ± 14.97 g m$^{-2}$ during the growing season (from May to September). The overall root biomass of all treatments showed an increasing trend and reached the maximum in September (data from May is not included in the figure because root biomass was considered to be dead for all samples taken in May). There was no significant difference in root biomass among all treatments in June, July or August. Root biomass of the SG treatment in September was significantly greater than that of the MG, CG and CK treatments, with *p*-values of 0.038, 0.001 and 0.001, respectively. Root biomass of the MG treatment in September was significantly greater that of the CK treatment ( *p* = 0.093).

The results of root biomass measured using the drilling method are shown in figure 2*b*. Total root biomass in each month on average ranged from 381.74 ± 43.28 to 813.28 ± 96.89 g m$^{-2}$ during the growth season. Overall, the root biomass of all treatments showed a steady or slight decrease over time. There was no significant difference in root biomass from May to August, but the root biomass of

royalsocietypublishing.org/journal/rsos　R. Soc. open sci. **6**: 180890

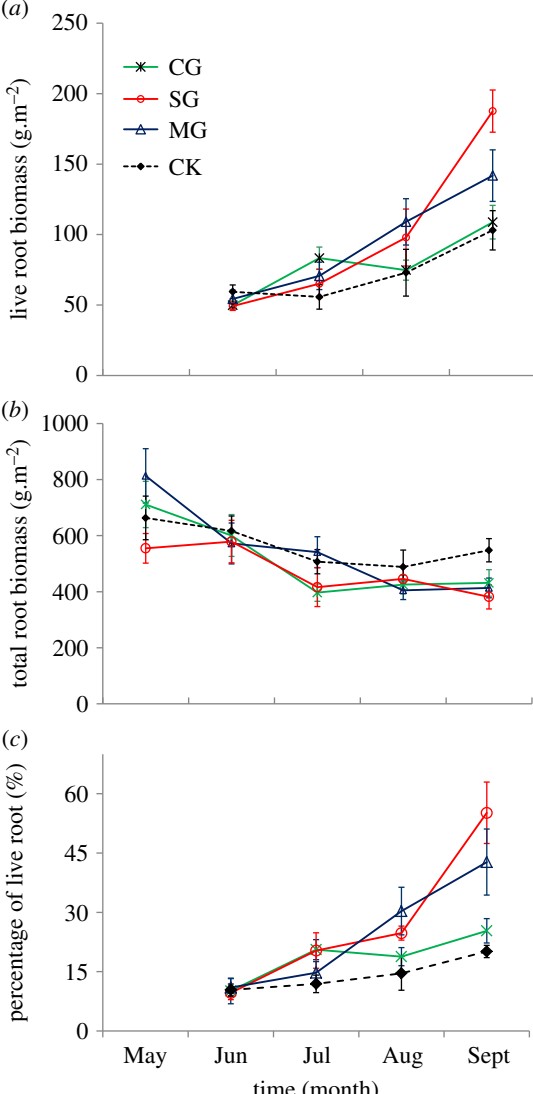

**Figure 2.** Variations in plant community root biomass in the 0 – 30 cm soil layer during the growing season in four large herbivore grazing treatments (CG, cattle grazing; SG, sheep grazing; MG, mixed grazing; CK, non-grazed control).

the CK treatment was significantly greater than that of the MG, SG and CG treatments in September, with $p$-values of 0.03, 0.08 and 0.056, respectively. There were no significant differences in root biomass between other treatments.

The ratio of live root biomass to total root biomass (figure 2$c$) increased with the time increase. The proportions of live roots in grazed areas were equal to or larger than that of the CK treatment. Live root biomass accounted for 7.43% $\pm$ 2.22% to 55.16% $\pm$ 8.24% of the total root system in the four treatments. There were no differences in the proportion of live root biomass between treatments in May and August, but the proportion of live roots was smaller than in later months. In September, the proportion of live roots in total root biomass in the SG treatment was significantly greater than in CG and CK ($p = 0.028$ and 0.011, respectively).

## 3.2. Belowground net primary production and root turnover

BNPP and RTR were calculated using three methods during the growing season from May to September (table 1 and table 2). BNPP varied from 165.78 $\pm$ 32.06 to 500.65 $\pm$ 88.48 g m$^{-2}$ among all the treatments. For BNPP sampled using the drilling method and calculated by Method I and Method II, both methods showed that BNPP was the smallest in CG and was the largest in SG. The results of Method I showed that BNPP in MG, SG and CK were significantly greater than in CG ($p = 0.029$, 0.017 and 0.098, respectively). The results of Method II showed that BNPP in CG was significantly smaller than in SG and CK ($p = 0.08$

**Table 1.** BNPP of the plant community in four large herbivore grazing treatments (CG, cattle grazing; SG, sheep grazing; MG, mixed grazing; CK, non-grazed control), as calculated using three methods. Significant differences between treatments are indicated by different letters, mean $\pm$ s.e. Method I—summation of positive increases in total root biomass; Method II—summation of significant positive increases in root biomass by depth and Method III—summation of positive increases in live root biomass.

| treatments | BNPP (g m$^{-2}$ year$^{-1}$) | | |
|---|---|---|---|
| | Method I | Method II | Method III |
| CG | 217.40 $\pm$ 64.47B[a] | 165.78 $\pm$ 32.06C | 284.50 $\pm$ 18.06b |
| SG | 500.65 $\pm$ 88.48A | 437.90 $\pm$ 69.96AB | 381.35 $\pm$ 31.08a |
| MG | 474.64 $\pm$ 113.04A | 288.96 $\pm$ 28.98BC | 286.54 $\pm$ 6.29b |
| CK | 397.12 $\pm$ 34.82A | 400.14 $\pm$ 22.32A | 300.22 $\pm$ 22.02b |

[a]Capital letters and lower case letters indicate significance levels for $\alpha$ of 0.1 and 0.05, respectively.

**Table 2.** Root turnover rate per year in a steppe grassland plant community in four large herbivore grazing treatments (CG, cattle grazing; SG, sheep grazing; MG, mixed grazing; CK, non-grazed control), as calculated using three methods. Significant differences between treatments are indicated by different lower case letters, mean $\pm$ s.e. Method I—summation of positive increases in total root biomass; Method II—summation of significant positive increases in root biomass by depth and Method III—summation of positive increases in live root biomass.

| treatments | root turnover rate (year$^{-1}$) | | |
|---|---|---|---|
| | Method I | Method II | Method III |
| CG | 0.29 $\pm$ 0.09b | 0.25 $\pm$ 0.05c | 0.45 $\pm$ 0.03a |
| SG | 0.70 $\pm$ 0.11a | 0.68 $\pm$ 0.12a | 0.58 $\pm$ 0.08a |
| MG | 0.62 $\pm$ 0.10a | 0.39 $\pm$ 0.07bc | 0.43 $\pm$ 0.04a |
| CK | 0.53 $\pm$ 0.06a | 0.53 $\pm$ 0.04ab | 0.38 $\pm$ 0.05a |

and 0.002, respectively) and that MG was significantly smaller than CK ($p = 0.09$). The results of Method III were similar to those of Methods I and II, with BNPP in CG, MG and CK smaller than in the SG treatment ($p = 0.017$, 0.023 and 0.029, respectively).

RTR varied from 0.25 to 0.70 year$^{-1}$ among the four treatments. Results of Methods I and II both showed that RTR was larger in SG and smaller in the CG treatments. The turnover rates of MG, SG and CK were significantly greater than that of CG when calculated by Method I ($p = 0.017$, 0.004 and 0.065, respectively). When calculated using Method II, the turnover rates of CG and MG were significantly smaller than that of SG ($p = 0.001$ and 0.011, respectively), while CG was significantly smaller than CK ($p = 0.01$). Method III showed no significant differences among the turnover rates of all treatments.

We also analysed the correlation between the ratio of the live root mass to total root mass and the root turnover rate calculated by the three methods (figure 3). Results show a positive correlation curve among the four treatments based on Methods I and III. A similar relationship was also obtained when using Method II, but the relationship was not statistically significant.

## 3.3. Standing live biomass of functional groups and aboveground biomass of the community

*Stipa krylovii* and *Leymus chinensis* were the dominant species of the plant community. The standing live biomass of *Stipa krylovii* ranged from 8.83 $\pm$ 1.16 to 13.13 $\pm$ 3.01 g m$^{-2}$ (figure 4) and there were no significant differences between the treatments. The standing live biomass of *Leymus chinensis* ranged from 13.58 $\pm$ 1.50 to 25.20 $\pm$ 3.83 g m$^{-2}$. The standing live biomass of *Leymus chinensis* in the CG treatment was significantly smaller than that in the CK treatment. The standing live biomass of Gramineae species ranged from 18.42 $\pm$ 1.72 to 37.98 $\pm$ 3.34 g m$^{-2}$ and in the CK treatments was greatest among the treatments. The standing live biomass of *Leguminosae* ranged from 3.74 $\pm$ 0.74 g m$^{-2}$ to 5.61 $\pm$ 1.14 g m$^{-2}$.

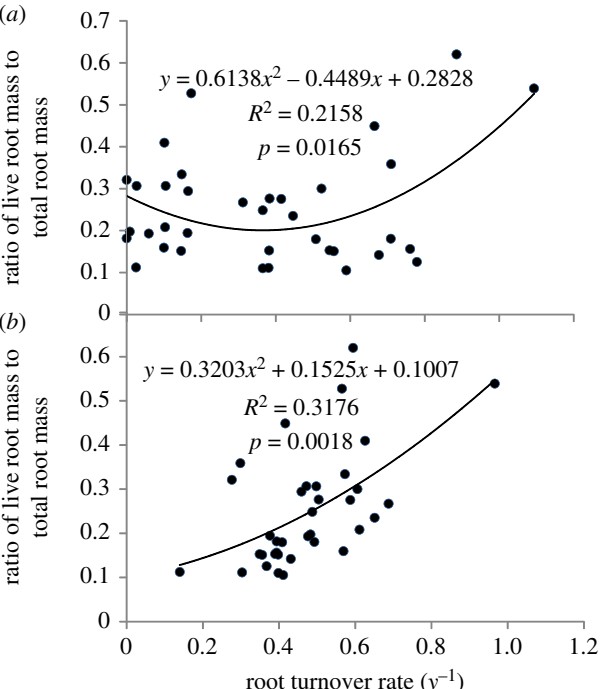

**Figure 3.** The relationships between root turnover rate and the ratio of live root biomass to total root biomass in four herbivore grazing treatments (*a*—based on data calculated using Method I; *b*—based on data calculated using Method III).

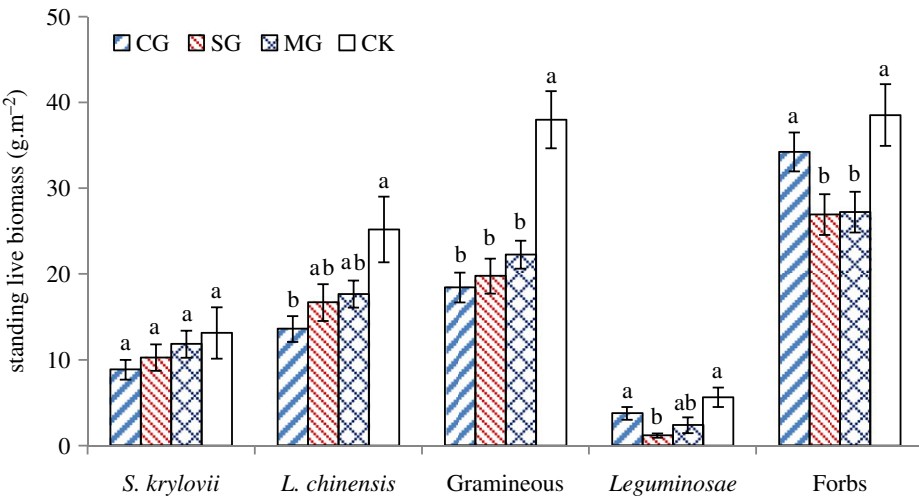

**Figure 4.** Standing live biomass of functional groups in the plant community in four herbivore grazing treatments (CG, cattle grazing; SG, sheep grazing; MG, mixed grazing; CK, non-grazed control). Significant differences between treatments in the same group are indicated by different lower case letters.

The standing live biomass of *Leguminosae* in the SG treatment was significantly smaller than that in the CK treatment and CG treatments. The standing live biomass of *Leguminosae* in MG treatment was in-between the CG and SG treatment and there were no significant differences among these three treatments. The standing live biomass of forbs ranged from $34.22 \pm 2.27$ to $38.53 \pm 3.60$ g m$^{-2}$ and was significantly smaller in SG and MG treatment than in CK treatment and CG treatments, while there were no differences between the SG and MG treatments and between the CK and CG treatments.

Aboveground biomass reached a maximum in August (figure 5*b*). Aboveground biomass in the control (CK) treatment was significantly greater than that in other treatments from June to September ($p < 0.01$). The biomass of the SG treatment was smaller than that of the MG and CG treatments before August ($p < 0.1$), while the biomass of the CG treatment was also smaller than that of the MG and SG treatments in September ($p < 0.01$). There were no significant differences in the biomass of the other treatments.

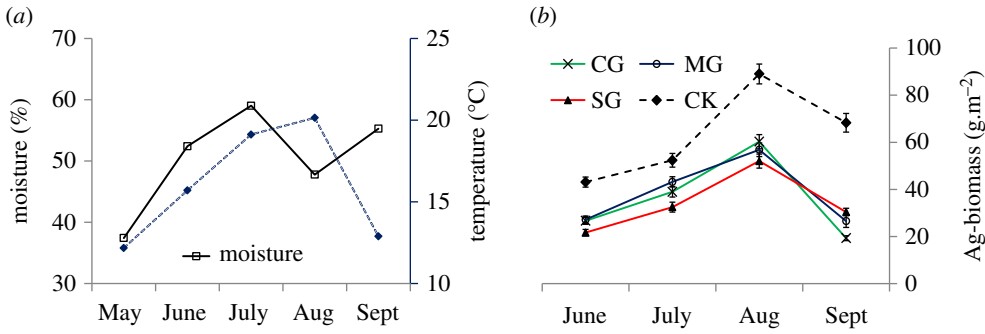

**Figure 5.** Environmental moisture, temperature (*a*), and aboveground biomass (*b*) at the experiment site in 2016.

# 4. Discussion

## 4.1. Distribution of root biomass in the community during the growing season

In this study, total root biomass showed a decreasing trend while live root biomass increased with time in the growing season in this study. The root biomass obtained using the drilling method includes live and dead roots in this study. Change in this variable is related to the growth of live roots and the decomposition rate of dead roots. While the growth of living roots is related to water availability and temperature, moisture and heat are the important constraining factors on plant growth in arid and semiarid areas [28]. Meteorological data for the experimental site show that soil moisture in the growing season was at a maximum in July after which it decreased in August (figure 5*a*). Drought could stimulate root growth to obtain more resources. As rainfall was abundant in June and July, total root biomass was low (figure 2*b*) while photosynthate was allocated much more to shoot growth (figure 5*b*). The decomposition of the dead root is related to root characteristics and decomposer communities [29]. Roots are decomposed faster in the upper soil than in deeper soil of the SG treatment [30,31]. In this study, in the SG treatment, a greater proportion of roots occurred more in the upper soil compared to the other treatments and there was a smaller root mass in deeper soils of the SG treatment. As a result, the proportion of live roots in total root mass was the largest in the SG treatment in this study. The increment of root biomass at different stages measured by the ingrowth method reflected the change in root growth rate and indicated that plants accelerated root growth in the growing season. Aboveground biomass was at a maximum in August, while the root biomass maximum was in September. Feng and Chai *et al.* [32,33] also reported that the root biomass maximum occurs one month later than the aboveground biomass, and plants allocate more photosynthate to belowground parts at the end of the growing season in order to survive in the cold winter. The temperature falls quickly in October in the study region at which point all aboveground plants die.

The root biomass in the soil profile decreased with the soil depth. However, more root tended to occur in surface soil after grazing, as was also found by other studies [34,35].

In this study, the proportion of living roots in total root biomass was between 7.43% ± 2.22% and 55.16% ± 8.24% from May to September, and the proportion in grazing areas was equal to or higher than that of no-grazing areas. Gao *et al.* [14] reported the proportion was in-between 22% and 36% (mean live root fraction in 2 years), and that the ratio was smaller in no-grazing areas in a typical steppe in Inner Mongolia [14]. Changes in this ratio are related to increased growth of live roots and decomposition of the dead root, which should vary by month and location.

## 4.2. The impacts of grazing on root biomass, BNPP and root turnover

Based on the three methods, the range of BNPP for no-grazing areas was 300–400 g m$^{-2}$. Hou *et al.* [36] reported that BNPP was 450.2 g m$^{-2}$ (equal to 392 g m$^{-2}$ if calculated with an ash content of 12.85%) for a non-grazed area of desert steppe in Su'niteyou County, Inner Mongolia. In addition, the results obtained by Gao *et al.* [14] in a non-grazed area of a typical steppe in Xilinghaote, Inner Mongolia were 366–464 g m$^{-2}$ while results obtained by the ingrowth method were 180–203 g m$^{-2}$ year$^{-1}$. BNPP was 300 g m$^{-2}$ year$^{-1}$ in our study, which is smaller than results for a typical steppe, and may

be caused by different hydrothermal conditions in the study area. Long-term average precipitation is 343 mm in a typical steppe, while it is 284 mm in a desert steppe in our study. Root turnover rate calculated by the three methods ranged from 0.25 to 0.70 year$^{-1}$ with a soil depth of 30 cm in this study. In a typical steppe in Inner Mongolia, root turnover rate ranged from 0.23 to 0.33 year$^{-1}$ with a soil depth of 100 cm, where the dominant species were *Leymus chinensis* and *Stipa grandis* [14]. López-Mársico [37] found that root turnover of a temperate grassland in central-southern Uruguay was 0.4 year$^{-1}$ in grazed areas and 0.37 year$^{-1}$ in non-grazed areas. Sims & Singh [3] reported a root turnover rate of 0.49 year$^{-1}$ for shortgrass prairies in western North American grasslands. Chen *et al.* [38] also reported a root turnover rate of 0.49 year$^{-1}$ for typical steppe grasslands in Inner Mongolia. All the above results were obtained using the soil core methods. Different methods have different errors, especially when compared with traditional methods. For example, Bai *et al.* [39] estimated RTR of 2.17, 1.89 and 1.36 year$^{-1}$ for *S. krylovii*, *S. grandis* and *S. breviflora* grasslands, respectively, in Inner Mongolia using the minirhizotron method. The above results indicate that RTR obtained using different methods differ substantially, but comparisons can be made between the results obtained using similar methods.

We found that CG decreased root biomass and BNPP while SG increased the root biomass and BNPP. Factors affecting BNPP include water, temperature and vegetation types [40]. The main factor considered in this study was the impact of animal herbivory. Considering the feeding characteristics of livestock, cattle mainly feed on *Stipa krylovii*, *Leymus chinensis* and other Gramineae (figure 4), which are two of the iconic species in grassland on the Eurasian steppe and the dominant species in the steppe vegetation community, accounting for about 45–70% of the standing live biomass [38,41]. Cattle feed much less on legumes and forbs, which reduces the competition for Gramineae and leguminous forages or forbs as resources. A similar study in a typical steppe of Inner Mongolia also found that cattle ate mainly a high proportion of *Stipa grandis*, while sheep did not consume this species under light and moderate grazing intensity [42]. *Artemisia frigida* Willd. and *Cleistogenes songorica* Ohwi are the main dietary components for sheep. Sheep rarely choose *Stipa breviflora* Griseb or *S. krylovii*, especially in summer (from June to August) [43,44]. Thus, cattle and sheep differ in their feeding behaviours on the steppe. Large herbivores can tolerate plants with a lower nutrient content but require a greater abundance of their preferred plants. By contrast, small herbivores prefer to eat species with a higher nutrient content and smaller live biomass [45]. Sheep mainly eat delicate legumes and forbs, so sheep grazing increases the competitive advantage of *S. krylovii* and other Gramineae. This was supported by our results indicating that SG and CG decreased community diversity, but MG improved community diversity [46]. Liu *et al.* [23] found that CG and MG also decreased plant biomass under moderate grazing intensity in a low plant diversity steppe grassland [23]. The desert steppe in this study is the driest grassland type in Inner Mongolia and is also a low diversity grassland. The grassland in the CG treatment had the smallest aboveground biomass in September (figure 5*b*). Root biomass in the CG treatment would decrease as the aboveground biomass decreased, as this would make fewer photosynthetic products available to be transported to belowground plant parts.

Factors affecting root turnover and longevity include temperature, the availability of water and nutrients, mycorrhizal fungi and disturbances from herbivores, pathogens and microbes. Root turnover and longevity also differ among different plant species and plant types (e.g. coarse versus fine rooted, obligate versus facultative mycotrophs) [38]. The root turnover rate was higher in SG and lower in CG, as determined by all three methods in this study except for the turnover rate calculated using Method III (no significant differences, sig. = 0.106). Legumes and forbs are dicotyledonous plants and usually their roots are relatively sparser and deeper than the roots of Gramineae, which are monocotyledonous with a fibrous root system [47,48]. In general, the roots of Gramineae are dense, while roots of legumes and forbs are coarse with a taproot and a swollen junction (root crown) or bulb that usually contains abundant carbohydrates. So roots of many legumes and forbs are thicker than Gramineae within 0–30 cm soils. This means that they have a greater longevity and a slower turnover rate as root turnover is mainly related to root diameter and the amount of soluble sugars in the root system [39,49,50]. Roots with higher sugar content and large diameters have a longer lifespan and slower turnover while a high-specific root length (ratio of root length to dry mass) often indicates a short lifespan [51].

In this study, there was a positive correlation among the four treatments between the ratio of live root biomass to total root biomass and the root turnover rate among the four treatments when measured using Methods I and III (figure 5). The same correlation was also found by Fial [52] based on the study in meadow stands of the Czech Republic.

# 5. Conclusion

In conclusion, our results demonstrated that the total root biomass of the community showed a slightly decreasing trend while live root biomass increased with time during the growing season. Herbivore assembly had different effects on root biomass and root turnover depending on the species. SG had the greatest root biomass, BNPP and root turnover rate, while the CG treatment had the smallest root biomass, BNPP and turnover rate. The results for the MG treatment were between those for the CG and SG treatments. Results for CK and MG were similar when estimated using two of the three calculations. The differences among the four treatments can be attributed to changes in the aboveground biomass of functional groups caused by animal herbivores. These findings based on root growth conditions after 2 years of experimental treatment in this study suggest that mixed grazing by cattle and sheep is associated with more stable and sustainable plant communities than CG or SG alone.

Ethics. The animal experiments complied with the current Chinese laws and were approved by the Experimental Animal Committee of the Chinese Academic of Science.

Data accessibility. Data available from the Dryad Digital Repository: https://doi.org/10.5061/dryad.dr357sv [53].

Authors' contributions. C.J.W. and G.D.H. designed and performed the experiments, and revised the manuscript; Z.Y.W. performed the experiments, analysed the data and wrote the manuscript; J.J. and Y.N.Z., Y.L.J., X.J.L. performed the experiments and contributed to data analysis. All authors gave final approval for publication.

Competing interests. The authors declare that they have no competing interests.

Funding. This research is funded by two grants from the National Natural Science Foundation (NNSF) of China. Z.Y.W. and C.J.W. were supported by NNSF with grant no. 31560141 and 31460125, respectively. This research is partly funded by a National Key Research and Development Program grant (grant no. 2016YFC0500504 provided to G.D.H.) from the Ministry of Science and Technology of the People's Republic of China.

Acknowledgements. We particularly appreciate the assistance in the fieldwork by undergraduate students: Nan Xiao, Xuejiao Duan and Jichao Wang and by a farmer: Jianjun Du. This research was partially supported by the Innovation, Research Team of Ministry of Education (IRT17R59), China Scholarship Council (CSC) scholarships and Inner Mongolia Key Laboratory of Grassland Management and Utilization. Thanks to Professor Cory Matthew and Ignacio F. López of Massey University for their valuable suggestions during the revision of manuscripts and to Louise Brok, Massey University postgraduate student, for her assistance with proof reading.

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
