## [Reviewer comments · Royal Society Open Science]

Review History

RSOS-180890.R0 (Original submission)

Review form: Reviewer 1

Is the manuscript scientifically sound in its present form?

Yes

Are the interpretations and conclusions justified by the results?

No

Is the language acceptable?

No

Is it clear how to access all supporting data?

Yes

Do you have any ethical concerns with this paper?

No

Have you any concerns about statistical analyses in this paper?

No

Recommendation?

Major revision is needed (please make suggestions in comments)

Comments to the Author(s)

The field experiment was carried out in Xilamuren desert steppe, China. Root biomass, belowground net primary production (BNPP) and plant community were measured with four treatments: sheep grazing alone (SG), cattle grazing (CG), mixed grazing with cattle and sheep (MG), no grazing (CK). SG had the highest BNPP and root turnover rate, while the CG had the lowest BNPP and root turnover rate. Compared with other treatments, CG had a greater impact on dominant tall grasses species in communities. SG could decrease community diversity. MG eliminated the disadvantages of single-species grazing and was beneficial to community diversity. And in conclusion, MG could lead to more sustainable grassland use. This paper give a very interesting story, however, the writing is bad and the authors not clearly and logically present the tables and figures and also language. Therefore the manuscript needs a major revision to meet the goal of the journal of Royal Society Open Science.

1. The study goal in this manuscript is not very clear.
2. The summary section didn't describe the main methods, results and conclusion logically and clearly.
3. The figures and tables need to be modified and the normative need to be improved.
4. Deeper and logical discussion is needed to be shown in this manuscript.
5. English scientific writing in each part of manuscript needs to be enhanced. A lot grammar, temporal, space problems are existed in this manuscript.

The followings are suggestions in detail.

Summary

The results of root turnover and BNPP under four grazing treatments were shown in the first half part of summary, then plant community and diversity were shown. How grazing treatment affect root turnover and BNPP? What is the relationship between root turnover, BNPP and plant community, and the sustainable grassland use?

"Few have explored..." should be "Few studies have explored..."

"SG had a larger BNPP and root turnover rate, while the CG had smaller BNPP and root turnover rate." The comparative on BNPP and root turnover rate here were not clear. Here the superlative form should be used.

"The MG treatment was between the CG and SG treatments. CK and MG were similar among all the treatments in two of three methods". These sentences are lacked of the real subject, and what author want to explain is not clear.

Introduction

"...by altering plant species composition and affecting aboveground NPP" what is NPP? It should be presented in manuscript.

"These advantages may be due to cattle and sheep having different preferences for plants both with respect to species and plant parts selected during grazing, and a reduction in sward rejection due to dung contamination". The reference of these views should be given here.

Materials and methods

Overall, the "materials and methods" part is not specific enough.

For the part "Experimental design", the grazing years (2014 and 2015?) need to be given in this part.

"Root biomass was monitored by drilling holes and collecting and sampling the soil recovered, while root growth was measured by root ingrowth cores...The root biomass sampling sites were about 1 m away from the root ingrowth cores in each case" At the beginning of results part, "The following is the results for the root biomass assessed by the ingrowth method". What ingrowth

method was used to assess to, root biomass or root growth? As for the time of root sampling, were drilling holes and root ingrowth cores carried out at the same time (each month of the growing season)?

“Samples were separated at collection into three soil layers: three layers: 0–10 cm, 10–20 cm and 20–30 cm soil depth” The author should check the whole manuscript to avoid repeating statements, or unscientific writing.

“At least 22 plots, 0.5 m × 0.5 m quadrat, were sampled diagonally across each block... Biomass was assessed in twelve 0.5 m × 0.5 m plots in each block”. How many quadrats were investigated in each fenced paddock?

How live root biomass, total root biomass, BNPP and root turnover were measured should be also given detailed in part “materials and methods”, rather than be described in part “results”. Significant level should be given in statistical analysis part.

Results

The logic of the results description needs to be improved. The figures and tables of the results need to be modified and the normative need to be improved. It is suggested that figure 2 and 3 were also shown by “a” and “b” (i.e. Fig. 2a, 2b; Fig. 3a, 3b) to help reader to understand easily. Table 1 is suggested to change to figure and merge into figure 2. In table 2 the different letters for Methods II need to be check, since 437.90 was bigger than 400.14. Besides, what is difference between lower case letters and capital letters? The scale of Y axis in Figure 3a is suggested to be denser.

In part “Live root biomass and total root biomass”, “There was no significant difference in root biomass among all the treatments in June, July and August... Root biomass of the MG treatment was significantly greater than the CK ($p=0.093$).” Root biomass of the MG treatment in which month was significantly greater than that of the CK?

Fig. 4 and Fig. 5 showed stand live biomass and the relationships between root turnover rate and the ratio of live root biomass. It is suggested that these relationships are also described in results part of manuscript.

Discussion

The goal of this study is not clear. At the end of “Introduction” part, “Therefore, the goals of this research were: from the field experiments, to characterize the distribution of root biomass during a growing season in Inner Mongolian steppe grassland;...” but this goal is not shown in “summary” part at all. According to the title of this manuscript, the focus of this study should be exploring the impacts of grazing on root biomass and BNPP. However, these are not fully discussed in “Discussion” part.

The two subtitles of this part are lengthy or too simple. The whole discussion part lack of casual relationship, and structure needs to be adjusted.

“Aboveground biomass was measured by harvest method (Fig. 3b), and reached the maximum in August... However, more root tended to occurred in surface soil after grazing (see appendix S1 in supplements), which was the same finding as other studies”. “We also analysed the correlation between the ratio of live root mass to total root mass and the root turnover rate calculated by three methods... The same correlation was also found by Fial [47] based on the study in meadow stands of Czech Republic” These parts describe results, rather than discussion. Deeper discussion is needed to be shown and explore how stem and root parameters are affected by season or grazing.

“Legumes and forbs usually have relatively few roots, and their roots are thick. This means that they have greater longevity and a slower turnover rate.” “Generally, the roots of Gramineae grasses are fine and intensively dispersed”. The reference of these views should be given in these parts.

“In this study, the cattle mainly ate *S. krylovii* and *Leymus chinensis*, which are two of the iconic grassland species... In contrast, small herbivores prefer to eat species with a higher nutrient content and smaller live biomass”. This paragraph shown how is cattle feeding characteristics, and how cattle grazing affect vegetation. While how these feeding characteristics affect BNPP and root turnover?

Conclusions

The authors need to make conclusion according to the goals of this study. How herbivores grazing affect root biomass, BNPP and turnover? The author should also conclude it here.

Appendix

There are seven supporting information in appendix part. Only "appendix S1" can be found in manuscript (discussion part). How is other appendix information support this study? All of them need to be shown in this manuscript.

The appendix information should be simplified properly. Lots parameters in those supporting figures or tables were lack of unit.

References

The format needs to be check carefully according to journal's requirement. In this manuscript, some journal's titles are abbreviations, some titles are in full name.

Decision letter (RSOS-180890.R0)

05-Nov-2018

Dear Mr Wang,

The editors assigned to your paper ("Impacts of Mixed-grazing on Root Biomass and Belowground Net Primary Production on the Temperate Desert Steppe") have now received comments from reviewers. We would like you to revise your paper in accordance with the referee and Associate Editor suggestions which can be found below (not including confidential reports to the Editor). Please note this decision does not guarantee eventual acceptance.

Please submit a copy of your revised paper before 28-Nov-2018. Please note that the revision deadline will expire at 00.00am on this date. If we do not hear from you within this time then it will be assumed that the paper has been withdrawn. In exceptional circumstances, extensions may be possible if agreed with the Editorial Office in advance. We do not allow multiple rounds of revision so we urge you to make every effort to fully address all of the comments at this stage. If deemed necessary by the Editors, your manuscript will be sent back to one or more of the original reviewers for assessment. If the original reviewers are not available, we may invite new reviewers.

- Data accessibility

If you wish to submit your supporting data or code to Dryad (<http://datadryad.org/>), or modify your current submission to dryad, please use the following link:
<http://datadryad.org/submit?journalID=RSOS&manu=RSOS-180890>

- Competing interests

- Authors' contributions

- Acknowledgements

- Funding statement

Please note that Royal Society Open Science charge article processing charges for all new submissions that are accepted for publication. Charges will also apply to papers transferred to Royal Society Open Science from other Royal Society Publishing journals, as well as papers submitted as part of our collaboration with the Royal Society of Chemistry (<http://rsos.royalsocietypublishing.org/chemistry>). If your manuscript is newly submitted and

subsequently accepted for publication, you will be asked to pay the article processing charge, unless you request a waiver and this is approved by Royal Society Publishing. You can find out more about the charges at <http://rsos.royalsocietypublishing.org/page/charges>. Should you have any queries, please contact openscience@royalsociety.org.

on behalf of Prof. Jon Blundy (Subject Editor)
openscience@royalsociety.org

Associate Editor's comments:

We apologise for the delay in sending this decision: unfortunately, despite inviting nearly 20 reviewers, only one referee was prepared to assess your manuscript. To avoid delaying you further, the Editors have opted to make a decision based on this commentary. Please ensure that you fully respond to and incorporate the changes requested by this reviewer, as they will be invited to re-assess your revision. If they are not satisfied by the changes made, your paper may be rejected.

Comments to Author:

Reviewers' Comments to Author:

Reviewer: 1

Comments to the Author(s)

The field experiment was carried out in Xilamuren desert steppe, China. Root biomass, belowground net primary production (BNPP) and plant community were measured with four treatments: sheep grazing alone (SG), cattle grazing (CG), mixed grazing with cattle and sheep (MG), no grazing (CK). SG had the highest BNPP and root turnover rate, while the CG had the lowest BNPP and root turnover rate. Compared with other treatments, CG had a greater impact on dominant tall grasses species in communities. SG could decrease community diversity. MG eliminated the disadvantages of single-species grazing and was beneficial to community diversity. And in conclusion, MG could lead to more sustainable grassland use. This paper give a very interesting story, however, the writing is bad and the authors not clearly and logically present the tables and figures and also language. Therefore the manuscript needs a major revision to meet the goal of the journal of Royal Society Open Science.

1. The study goal in this manuscript is not very clear.
2. The summary section didn't describe the main methods, results and conclusion logically and clearly.
3. The figures and tables need to be modified and the normative need to be improved.
4. Deeper and logical discussion is needed to be shown in this manuscript.
5. English scientific writing in each part of manuscript needs to be enhanced. A lot grammar, temporal, space problems are existed in this manuscript.

The followings are suggestions in detail.

Summary

The results of root turnover and BNPP under four grazing treatments were shown in the first half part of summary, then plant community and diversity were shown. How grazing treatment affect

root turnover and BNPP? What is the relationship between root turnover, BNPP and plant community, and the sustainable grassland use?

“Few have explored...” should be “Few studies have explored...”

“SG had a larger BNPP and root turnover rate, while the CG had smaller BNPP and root turnover rate.” The comparative on BNPP and root turnover rate here were not clear. Here the superlative form should be used.

“The MG treatment was between the CG and SG treatments. CK and MG were similar among all the treatments in two of three methods”. These sentences are lacked of the real subject, and what author want to explain is not clear.

Introduction

“...by altering plant species composition and affecting aboveground NPP” what is NPP? It should be presented in manuscript.

“These advantages may be due to cattle and sheep having different preferences for plants both with respect to species and plant parts selected during grazing, and a reduction in sward rejection due to dung contamination”. The reference of these views should be given here.

Materials and methods

Overall, the “materials and methods” part is not specific enough.

For the part “Experimental design”, the grazing years (2014 and 2015?) need to be given in this part.

“Root biomass was monitored by drilling holes and collecting and sampling the soil recovered, while root growth was measured by root ingrowth cores...The root biomass sampling sites were about 1 m away from the root ingrowth cores in each case” At the beginning of results part, “The following is the results for the root biomass assessed by the ingrowth method”. What ingrowth method was used to assess to, root biomass or root growth? As for the time of root sampling, were drilling holes and root ingrowth cores carried out at the same time (each month of the growing season)?

“Samples were separated at collection into three soil layers: three layers: 0-10 cm, 10-20 cm and 20-30 cm soil depth” The author should check the whole manuscript to avoid repeating statements, or unscientific writing.

“At least 22 plots, 0.5 m ×0.5 m quadrat, were sampled diagonally across each block...Biomass was assessed in twelve 0.5 m ×0.5 m plots in each block”. How many quadrats were investigated in each fenced paddock?

How live root biomass, total root biomass, BNPP and root turnover were measured should be also given detailed in part “materials and methods”, rather than be described in part “results”. Significant level should be given in statistical analysis part.

Results

The logic of the results description needs to be improved. The figures and tables of the results need to be modified and the normative need to be improved. It is suggested that figure 2 and 3 were also shown by “a” and “b” (i.e. Fig. 2a, 2b; Fig.3a, 3b) to help reader to understand easily. Table 1 is suggested to change to figure and merge into figure 2. In table 2 the different letters for Methods II need to be check, since 437.90 was bigger than 400.14. Besides, what is difference between lower case letters and capital letters? The scale of Y axis in Figure 3a is suggested to be denser.

In part “Live root biomass and total root biomass”, “There was no significant difference in root biomass among all the treatments in June, July and August...Root biomass of the MG treatment was significantly greater than the CK ($p=0.093$).” Root biomass of the MG treatment in which month was significantly greater than that of the CK?

Fig. 4 and Fig. 5 showed stand live biomass and the relationships between root turnover rate and the ratio of live root biomass. It is suggested that these relationships are also described in results part of manuscript.

Discussion

The goal of this study is not clear. At the end of “Introduction” part, “Therefore, the goals of this research were: from the field experiments, to characterize the distribution of root biomass during

a growing season in Inner Mongolian steppe grassland;..." but this goal is not shown in "summary" part at all. According to the title of this manuscript, the focus of this study should be exploring the impacts of grazing on root biomass and BNPP. However, these are not fully discussed in "Discussion" part.

The two subtitles of this part are lengthy or too simple. The whole discussion part lack of casual relationship, and structure needs to be adjusted.

"Aboveground biomass was measured by harvest method (Fig.3b), and reached the maximum in August.... However, more root tended to occurred in surface soil after grazing (see appendix S1 in supplements), which was the same finding as other studies". "We also analysed the correlation between the ratio of live root mass to total root mass and the root turnover rate calculated by three methods...The same correlation was also found by Fial [47] based on the study in meadow stands of Czech Republic" These parts describe results, rather than discussion. Deeper discussion is needed to be shown and explore how stem and root parameters are affected by season or grazing.

"Legumes and forbs usually have relatively few roots, and their roots are thick. This means that they have greater longevity and a slower turnover rate." "Generally, the roots of Gramineae grasses are fine and intensively dispersed". The reference of these views should be given in these parts.

"In this study, the cattle mainly ate *S. krylovii* and *Leymus chinensis*, which are two of the iconic grassland species...In contrast, small herbivores prefer to eat species with a higher nutrient content and smaller live biomass". This paragraph shown how is cattle feeding characteristics, and how cattle grazing affect vegetation. While how these feeding characteristics affect BNPP and root turnover?

Conclusions

The authors need to make conclusion according to the goals of this study. How herbivores grazing affect root biomass, BNPP and turnover? The author should also conclude it here.

Appendix

There are seven supporting information in appendix part. Only "appendix S1" can be found in manuscript (discussion part). How is other appendix information support this study? All of them need to be shown in this manuscript.

The appendix information should be simplified properly. Lots parameters in those supporting figures or tables were lack of unit.

References

The format needs to be check carefully according to journal's requirement. In this manuscript, some journal's titles are abbreviations, some titles are in full name.

Author's Response to Decision Letter for (RSOS-180890.R0)

See Appendix A.

RSOS-180890.R1 (Revision)

Review form: Reviewer 1

Is the manuscript scientifically sound in its present form?

Yes

Are the interpretations and conclusions justified by the results?

Yes

Is the language acceptable?

Yes

Is it clear how to access all supporting data?

Yes

Do you have any ethical concerns with this paper?

No

Have you any concerns about statistical analyses in this paper?

No

Recommendation?

Accept as is

Comments to the Author(s)

Finally, please check whether you put all figures and tables information in the proper place in the discussion part? also check whether all references you cite appear simultaneously in main text parts and references part.

Decision letter (RSOS-180890.R1)

17-Dec-2018

Dear Mr wang:

On behalf of the Editors, I am pleased to inform you that your Manuscript RSOS-180890.R1 entitled "Impacts of Mixed-Grazing on Root Biomass and Belowground Net Primary Production in a Temperate Desert Steppe" has been accepted for publication in Royal Society Open Science subject to minor revision in accordance with the referee suggestions. Please find the referees' comments at the end of this email.

The reviewers and Subject Editor have recommended publication, but also suggest some minor revisions to your manuscript. Therefore, I invite you to respond to the comments and revise your manuscript.

- **Ethics statement**

- **Data accessibility**

It is a condition of publication that all supporting data are made available either as supplementary information or preferably in a suitable permanent repository. The data accessibility section should state where the article's supporting data can be accessed. This section should also include details, where possible of where to access other relevant research materials

such as statistical tools, protocols, software etc can be accessed. If the data has been deposited in an external repository this section should list the database, accession number and link to the DOI for all data from the article that has been made publicly available. Data sets that have been deposited in an external repository and have a DOI should also be appropriately cited in the manuscript and included in the reference list.

If you wish to submit your supporting data or code to Dryad (<http://datadryad.org/>), or modify your current submission to dryad, please use the following link:
<http://datadryad.org/submit?journalID=RSOS&manu=RSOS-180890.R1>

- **Competing interests**

- **Authors' contributions**

- **Acknowledgements**

- **Funding statement**

Because the schedule for publication is very tight, it is a condition of publication that you submit the revised version of your manuscript before 26-Dec-2018. Please note that the revision deadline will expire at 00.00am on this date. If you do not think you will be able to meet this date please let me know immediately.

Please note that Royal Society Open Science charge article processing charges for all new submissions that are accepted for publication. Charges will also apply to papers transferred to Royal Society Open Science from other Royal Society Publishing journals, as well as papers submitted as part of our collaboration with the Royal Society of Chemistry (<http://rsos.royalsocietypublishing.org/chemistry>). If your manuscript is newly submitted and subsequently accepted for publication, you will be asked to pay the article processing charge, unless you request a waiver and this is approved by Royal Society Publishing. You can find out more about the charges at <http://rsos.royalsocietypublishing.org/page/charges>. Should you have any queries, please contact openscience@royalsociety.org.

on behalf of Prof Jon Blundy (Subject Editor)
openscience@royalsociety.org

Reviewer comments to Author:

Reviewer: 1

Comments to the Author(s)

Finally, please check whether you put all figures and tables information in the proper place in the discussion part? also check whether all references you cite appear simultaneously in main text parts and references part.

Author's Response to Decision Letter for (RSOS-180890.R1)

See Appendix B.

Decision letter (RSOS-180890.R2)

09-Jan-2019

Dear Mr wang,

I am pleased to inform you that your manuscript entitled "Impacts of mixed-grazing on root biomass and belowground net primary production in a temperate desert steppe" is now accepted for publication in Royal Society Open Science.

on behalf of Prof Jon Blundy (Subject Editor)
openscience@royalsociety.org

Appendix A

Reviewers' Comments and authors responses	The updated part in the manuscript
Reviewers' Comments to Author: Reviewer: 1 Comments to the Author(s) This paper give a very interesting story, however, the writing is bad and the authors not clearly and logically present the tables and figures and also language. Therefore the manuscript needs a major revision to meet the goal of the journal of Royal Society Open Science. 1. The study goal in this manuscript is not very clear. Authors: we have revised the study goal in the manuscript 2. The summary section didn't describe the main methods, results and conclusion logically and clearly. Authors: we have revised the summary in the manuscript. 3. The figures and tables need to be modified and the normative need to be improved. Authors: The figures and tables were modified in the manuscript. 4. Deeper and logical discussion is needed to be shown in this manuscript. Authors: we have revised the discussion in the manuscript. 5. English scientific writing in each part of manuscript needs to be enhanced. A lot grammar, temporal, space problems are existed in this	The goals of this research were: to characterize the distribution of root biomass during a growing season in an Inner Mongolian steppe grassland, and 2) to evaluate the effects of herbivores assembly on root biomass, BNPP and RTR, using two sampling methods.

manuscript.

Authors: The manuscript's was checked again by

The followings are suggestions in detail.

Authors: we have revised the study goal in the manuscript

Summary

The results of root turnover and BNPP under four grazing treatments were shown in the first half part of summary, then plant community and diversity were shown. How grazing treatment affect root turnover and BNPP? What is the relationship between root turnover, BNPP and plant community, and the sustainable grassland use?

Authors:

“Few have explored...” should be “Few studies have explored...”

Authors: we have changed it.

“SG had a larger BNPP and root turnover rate, while the CG had smaller BNPP and root turnover rate.” The comparative on BNPP and root turnover rate here were not clear. Here the superlative form should be used.

Authors: Yes, the superlative form was used.

“The MG treatment was between the CG and SG treatments. CK and MG were similar among all the treatments in two of three methods”. These

1. Summary

The impacts of large herbivores on plant communities differ depending on the plants and the herbivores. Few **studies** have explored how herbivores influence root biomass. Root growth of vegetation was studied in the field with four treatments: sheep grazing alone (SG), cattle grazing **alone** (CG), mixed grazing with cattle and sheep (MG) and no grazing (CK). **Live and total root biomasses were measured using the root ingrowth core and the drilling core, respectively.** After 2 years of grazing, **total root biomass showed a decreasing trend while live root biomass increased with time during the growing seasons.** Belowground net primary production (BNPP) **among the treatments** varied from 166 ± 32 to 501 ± 88 g.m⁻² and root turnover rates (**RTR**) varied from 0.25 ± 0.05 to 0.70 ± 0.11 year⁻¹. SG had **the greatest** BNPP and **RTR**, while the CG had the **smallest** BNPP and RTR. **BNPP and RTR** of the MG treatment were between **those of** the CG and SG treatments. **BNPP and RTR** of the CK were similar to MG treatment. Compared with other treatments, CG had a greater impact on dominant tall grasses species in communities. SG could decrease community diversity. MG eliminated the disadvantages of single-species grazing and was beneficial to community diversity **and stability.**

Herbivory directly and indirectly impacts on below-ground biomass by altering plant species composition and affecting aboveground **net primary productivity(NPP),the difference between carbon fixed by photosynthesis and carbon lost to autotrophic**

sentences are lacked of the real subject, and what author want to explain is not clear.

Authors: Yes, we changed it. The real subject is BNPP and root turnover rate.

Introduction

“...by altering plant species composition and affecting aboveground NPP” what is NPP? It should be presented in manuscript.

“These advantages may be due to cattle and sheep having different preferences for plants both with respect to species and plant parts selected during grazing, and a reduction in sward rejection due to dung contamination”. The reference of these views should be given here.

Authors: we have changed it. And the reference was added into the manuscript!

Materials and methods

Overall, the “materials and methods” part is not specific enough.

For the part “Experimental design”, the grazing years (2014 and 2015?) need to be given in this part.

Authors: The grazing experiment commenced in 2014 and the grazing years were 2014,2015 and 2016.

“Root biomass was monitored by drilling holes and collecting and sampling

respiration [11, 12].

These advantages may be due to cattle and sheep having different preferences for plants both with respect to species and plant parts selected during grazing, and a reduction in sward rejection due to dung contamination [25].

Experimental design

The grazing experiment commenced in 2014 and included four treatments: cattle grazing (CG, 3 cattle per plot), sheep grazing (SG, 15 sheep per plot), mixed-grazing of cattle and sheep (MG, 3 cattle and 15 sheep per plot), and no grazing (CK). The grazing years were 2014, 2015 and 2016.

Root sampling

Two methods were used to characterize root dynamics at the same time: drilling core and root ingrowth core. Live root biomass was measured using the root ingrowth core while total root biomass was measured using the drilling core.

In each paddock, there were six drilling core sites and three ingrowth core sites, with the sampling sites for drilling cores about 1 m away from the root ingrowth core sampling sites. Samples were separated at collection into three soil layers: 0–10 cm, 10–20 cm and 20–30 cm in depth.

Vegetation survey

the soil recovered, while root growth was measured by root ingrowth cores...The root biomass sampling sites were about 1 m away from the root ingrowth cores in each case” At the beginning of results part, “The following is the results for the root biomass assessed by the ingrowth method”. **What ingrowth method was used to assess to, root biomass or root growth?** As for the time of root sampling, were drilling holes and root ingrowth cores carried out at the same time **(each month of the growing season)?**

Authors: Yes, the two methods were carried out at the same time in each month of the growing season. We use both methods to assess the root biomass and root growth at the same time.

“Samples were separated at collection into three soil layers: three layers: 0–10 cm, 10–20 cm and 20–30 cm soil depth”. **The author should check the whole manuscript to avoid repeating statements, or unscientific writing.**

Authors: Sorry for this. We have deleted the repeating statements and checked the whole manuscript.

“At least 22 plots, 0.5 m ×0.5 m quadrat, were sampled diagonally across each block...Biomass was assessed in twelve 0.5 m ×0.5 m plots in each block”. How many quadrats were investigated in each fenced paddock?

Authors: In each fenced paddock, 20 quadrats for MG treatment and

Quadrats were sampled diagonally across each paddock to investigate vegetation growth in August 2016. In each fenced paddock, twenty 0.5 m ×0.5 m quadrats for the MG treatment and 16 quadrats for the other treatments were set for monitoring the aboveground biomass. All plants in the quadrats were cut to a stubble height of 1 cm. Parameters measured included vegetation cover, height (estimated using measurements on five to seven plants of each species), shoot biomass, and botanical composition of the community.

Statistical analysis

BNPP ($\text{g}\cdot\text{m}^{-2}\cdot\text{yr}^{-1}$) was estimated using three methods based on Sims and Singh [3]. The methods were: (I) summation of positive increases in total root biomass**(0–30 cm) obtained by** the drilling methods; (II) Summation of positive increases in root biomass by depth **obtained by** the drilling methods; and (III) summation of positive increases in live root biomass **obtained by the ingrowth core** method. Root turnover rates were calculated by dividing the annual increment **(BNPP)** by peak **total** root biomass [3].

All data analysis was carried out in SPSS software version 24.0 (SPSS Inc., Chicago, IL, USA). Least Significant Difference (LSD) and Tamhane’s T2 were used for multiple comparisons of significant differences among treatments. The **significance level was set as 0.1 as it was a field trial.**

Fig.2

16 quadrats for the rest of treatments were set for monitoring the aboveground biomass. Each treatment includes three paddocks or replicates, so there are 60 quadrats for MG treatment and 48 quadrats for each of other treatments.

How live root biomass, total root biomass, BNPP and root turnover were measured should be also given detailed in part “materials and methods”, rather than be described in part “results”.

Significant level should be given in statistical analysis part.

Authors: These parts have been updated!

Results

The logic of the results description needs to be improved. The figures and tables of the results need to be modified and the normative need to be improved. It is suggested that figure 2 and 3 were also shown by “a” and “b” (i.e. Fig. 2a, 2b; Fig.3a, 3b) to help reader to understand easily. Table 1 is suggested to change to figure and merge into figure 2. In table 2 the different letters for Methods II need to be check, since 437.90 was bigger than 400.14. Besides, what is difference between lower case letters and capital letters? The scale of Y axis in Figure 3a is suggested to be denser.

Authors: Figure 2 and 3 were also shown by “a” and “b”. Table 1 was changed into a figure and merge into figure 2.

Fig.5

We checked the data and analysed it again in table 2, there is no problem!

Capital letters and lower case letters indicate that significant level for α were set as 0.1 and 0.05, respectively. The scale of Y axis in Figure 3a was also changed and is shown in Fig.5a.

In part “Live root biomass and total root biomass”, “There was no significant difference in root biomass among all the treatments in June, July and August...Root biomass of the MG treatment was significantly greater than the CK (p=0.093).” Root biomass of the MG treatment in which month was significantly greater than that of the CK?

Authors: It was in September. We have updated it!

Fig. 4 and Fig. 5 showed stand live biomass and the relationships between root turnover rate and the ratio of live root biomass. It is suggested that these relationships are also described in results part of manuscript.

Authors: Yes, we have move fig.4 and fig.5 into results part and now fig.5 was renumbered and changed into fig.3.

Discussion

The goal of this study is not clear. At the end of “Introduction” part, “Therefore, the goals of this research were: from the field experiments, to

Root biomass of the MG treatment in September was significantly greater than that of the CK treatment (p=0.093).

Discussion

Distribution of root biomass in the community during the growing season

In this study, total root biomass showed a decreasing trend while live root biomass

characterize the distribution of root biomass during a growing season in Inner Mongolian steppe grassland;...” but this goal is not shown in “summary” part at all. According to the title of this manuscript, the focus of this study should be exploring the impacts of grazing on root biomass and BNPP. However, these are not fully discussed in “Discussion” part.

Authors: we have added the distribution of root biomass during a growing season into summary and revised the discussion based on the goal.

The two subtitles of this part are lengthy or too simple. The whole discussion part lack of casual relationship, and structure needs to be adjusted.

Authors: we have revised the two subtitles of discussion: 1) Distribution of root biomass in the community during a growing season; 2) The impacts of grazing on root biomass, BNPP and root turnover.

The structure of discussion was also revised.

“Aboveground biomass was measured by harvest method (Fig.3b), and reached the maximum in August.... However, more root tended to occurred in surface soil after grazing (see appendix S1 in supplements), which was the same finding as other studies”. “We also analysed the correlation between the ratio of live root mass to total root mass and the root turnover

increased with time in the growing season in this study. The root biomass obtained using the drilling method includes live and dead roots in this study. Change in this variable is related to the growth of live roots and the decomposition rate of dead roots. While the growth of living roots is related to water availability and temperature, moisture and heat are the important constraining factors on plant growth in arid and semiarid areas [28]. Meteorological data for the experimental site shows that, soil moisture in the growing season was at a maximum in July after which it decreased in August (Fig. 5a). Drought could stimulate root growth to obtain more resources. As rainfall was abundant in June and July, total root biomass was low (Fig. 2b) while photosynthate was allocated much more to shoot growth (Fig. 5b). The decomposition of dead root is related to root characteristics and decomposer communities [29]. Roots are decomposed faster in the upper soil than in deeper soil of the SG treatment [30, 31]. In this study, in the SG treatment, a greater proportion of roots occurred more in the upper soil compared to the other treatments and there was a smaller root mass in deeper soils of the SG treatment (see Appendix S1 in the supplements). As a result, the proportion of live roots in total root mass was largest in the SG treatment in this study. The increment of root biomass at different stages measured by the ingrowth method reflected change in root growth rate, and indicated that plants accelerated root growth in the growing season. Aboveground biomass was at a maximum in August, while the root biomass maximum was in September. Feng and Chai *et al.* [32,33] also reported that the root biomass maximum occurs one month later than the aboveground biomass, and plants allocate more photosynthate to belowground parts at the end of the growing season in order to survive in the cold winter. The temperature falls quickly in October in the study region at which point all plants aboveground die.

The root biomass in the soil profile decreased with the soil depth. However, more root tended to occurred in surface soil after grazing(see Appendix S1 in the supplements),

rate calculated by three methods...The same correlation was also found by Fial [47] based on the study in meadow stands of Czech Republic” These parts describe results, rather than discussion. Deeper discussion is needed to be shown and explore how stem and root parameters are affected by season or grazing.

Authors: Yes, we have moved it into the results part and revised the discussion part.

“Legumes and forbs usually have relatively few roots, and their roots are thick. This means that they have greater longevity and a slower turnover rate.” “Generally, the roots of Gramineae grasses are fine and intensively dispersed”. The reference of these views should be given in these parts.

Authors: the references were added into this part.

Legumes and forbs are dicotyledonous plants and usually have relatively sparser and deeper roots than roots of Gramineae grasses, which are monocotyledonous and have fibrous root system(Weaver,1958; Chen,2001). Root longevity was correlated positively with root diameter (Wu,2013). This means that they have greater longevity and a slower turnover rate.”

“In this study, the cattle mainly ate *S. krylovii* and *Leymus chinensis*, which are two of the iconic grassland species...In contrast, small herbivores

as was also found by other studies [34, 35].

In this study, the proportion of living roots in total root biomass was between $7.43\% \pm 2.22\%$ and $55.16\% \pm 8.24\%$ from May to September, and the proportion in grazing areas was equal to or higher or the same than that of no-grazing areas. Gao *et al.* (2008) reported the proportion was in-between 22% and 36% (mean live root fraction in two year), and that ratio was smaller in no-grazing areas in a typical steppe in Inner Mongolia [14]. Changes in this ratio are related to increased growth of live roots and decomposition of dead root, which should vary by month and location.

The impacts of grazing on root biomass, BNPP and root turnover

Based on the three methods, the range of BNPP for no-grazing areas was 300 g.m^{-2} to 400 g.m^{-2} . Hou *et al.* [36] reported that BNPP was 450.2 g.m^{-2} (equal to 392 g.m^{-2} if calculated with an ash content of 12.85%) for a non-grazed area of desert steppe in Su'niteyou County, Inner Mongolia. In addition, the results obtained by Gao *et al.* [14] in a non-grazed area of a typical steppe in Xilinghaote, Inner Mongolia were $366\text{--}464 \text{ g.m}^{-2}$ while results obtained by the ingrowth method were $180\text{--}203 \text{ g.m}^{-2} \text{ year}^{-1}$. BNPP was $300 \text{ g.m}^{-2} \text{ year}^{-1}$ in our study, which is smaller than results for typical steppe, and may be caused by different hydrothermal conditions in the study area. Long-term average precipitation is 343 mm in typical steppe, while it is 284 mm in desert steppe in our study. Root turnover rate calculated by the three methods ranged from $0.25\text{--}0.70 \text{ year}^{-1}$ with a soil depth of 30 cm in this study. In a typical steppe in Inner Mongolia, root turnover rate ranged from 0.23 to 0.33 year^{-1} with a soil depth of 100 cm, where the dominant species were *Leymus chinensis* and *Stipa grandis* [14]. López-Mársico L. [37] found that root turnover of a temperate grassland in central-southern Uruguay was 0.4 year^{-1} in grazed areas and 0.37 year^{-1} in non-grazed areas. Sims & Singh [3] reported a root turnover rate of 0.49 year^{-1} for shortgrass prairies in western North American grasslands. Chen *et al.*

prefer to eat species with a higher nutrient content and smaller live biomass”. This paragraph shown how is cattle feeding characteristics, and how cattle grazing affect vegetation. While how these feeding characteristics affect BNPP and root turnover?

Authors: we revised this part.

Conclusions

The authors need to make conclusion according to the goals of this study. How herbivores grazing affect root biomass, BNPP and turnover? The author should also conclude it here.

Authors:

Appendix

There are seven supporting information in appendix part. Only “appendix S1” can be found in manuscript (discussion part). How is other appendix information support this study? All of them need to be shown in this manuscript.

The appendix information should be simplified properly. Lots parameters in those supporting figures or tables were lack of unit.

Authors: Sorry, All the supporting information have been upload in an external repository-Dryad . please use the following link:

<https://datadryad.org/review?doi=doi:10.5061/dryad.22pp13b>

[38] also reported a root turnover rate of 0.49 year⁻¹ for typical steppe grasslands in Inner Mongolia. All above results were obtained using the soil core methods. Different methods have different errors, especially **when comparing** and traditional methods. For example, Bai *et al.* [39] estimated root turnover rates of 2.17, 1.89 and 1.36 year⁻¹ for *S. krylovii*, *S. grandis* and *S. breviflora* grasslands in Inner Mongolia using the minirhizotron method, respectively. The above results indicate that root turnover rates obtained using different methods differ substantially, but comparisons can be made between the results obtained using similar methods.

We found that CG decreased root biomass and BNPP while SG increased the root biomass and BNPP. Factors affecting BNPP include water, temperature, and vegetation types [40]. The main factor considered in this study was the impact of animal herbivory. Considering the feeding characteristics of livestock, **cattle mainly feed on *Stipa krylovii*, *Leymus chinensis* and other Gramineae (Fig. 4), which are two of the iconic species in grassland on the Eurasian steppe and the dominant species in the steppe vegetation community, accounting for about 45%–70% of the standing live biomass [38, 41]. Cattle feed much less on legumes and forbs, which reduce the competition between Gramineae and leguminous forages or forbs for resources. A similar study in a typical steppe of Inner Mongolia also found that cattle mainly ate a high proportion of *Stipa grandis*, while sheep did not consume this species under light and moderate grazing intensity[42]. *Artemisia frigida* Willd. and *Cleistogenes songorica* Ohwi are the main dietary components for sheep. Sheep rarely choose *Stipa breviflora* Griseb or *S. krylovii*, especially in summer (from June to August) [43, 44]. Thus, cattle and sheep differ in their feeding behavior on the steppe. Large herbivores can tolerate plants with a lower nutrient content but require greater abundance of their preferred plants. In contrast, small herbivores prefer to eat species with a higher nutrient content and smaller live biomass [45]. Sheep mainly eat delicate legumes and forbs, so sheep grazing increases the competitive advantage of *S.***

References

The format needs to be checked carefully according to journal's requirement. In this manuscript, some journal's titles are abbreviations, some titles are in full name.

Authors: Sorry, I believed in too much to the Endnote software system in the past. And now it is updated. All changes in references were marked in red.

krylovii and other Gramineae. This was supported by our results indicating that SG and CG decreased community diversity, but MG improved community diversity [46]. Liu et al. (2015) found that CG and MG also decreased plant biomass under moderate grazing intensity in a low plant diversity steppe grassland [23]. The desert steppe in this study is the driest grassland type in Inner Mongolia and is also a low diversity grassland. The grassland in the CG treatment had the smallest aboveground biomass in September (Fig.5b). Root biomass in the CG treatment would decrease as the aboveground biomass decreased, as this would make fewer photosynthetic products available to be transported to belowground plant parts.

Factors affecting root turnover and longevity include temperature, the availability of water and nutrients, mycorrhizal fungi, and disturbances from herbivores, pathogens, and microbes. Root turnover and longevity also differ among different plant species and plant types (e.g. coarse vs. fine rooted, obligate vs. facultative mycotrophs)[38]. The root turnover rate was higher in SG and lower in CG, as determined by all three methods in this study except for the turnover rate calculated using Method III (no significant differences, sig.=0.106). Legumes and forbs are dicotyledonous plants and usually their roots are relatively sparser and deeper than the roots of Gramineae, which are monocotyledonous with a fibrous root system[47,48]. In general, the roots of Gramineae are dense, while roots of legumes and forbs are coarse with a taproot and a swollen junction (root crown) or bulb that usually contains abundant carbohydrates. So roots of many legumes and forbs are thicker than Gramineae within 0–30 cm soils. This means that they have greater longevity and a slower turnover rate as root turnover is mainly related to root diameter and the amount of soluble sugars in the root system [39,49,50]. Roots with higher sugar content and large diameters have a longer lifespan and slower turnover while a high specific root length (ratio of root length to dry mass) often indicates a short lifespan [51].

In this study, there was a positive correlation among the four treatments between the

ratio of live root biomass to total root biomass and the root turnover rate among the four treatments when measured using Methods I and III (Fig.5). The same correlation was also found by Fial [52] based on the study in meadow stands of the Czech Republic .

Conclusion

In conclusion, our results demonstrated that the total root biomass of the community showed a slightly decreasing trend while live root biomass increased with time during the growing season. Herbivore assembly had different effects on root biomass and root turnover depending on the species. SG had a greatest root biomass, BNPP and root turnover rate, while the CG treatment had a smallest root biomass, BNPP and turnover rate. The results for the MG treatment were between those for the CG and SG treatments. Results for CK and MG were similar when estimated using two of the three calculation. The differences among the four treatments can be attributed to changes in the aboveground biomass of functional groups caused by animal herbivores. These findings based on root growth conditions after the two years of experimental treatment in this study suggest that mixed grazing by cattle and sheep are associated with more stable and sustainable plant communities than CG or SG alone.

References

All changes in references were marked in red.

Appendix B

3) Reviewer comments to Author:

Reviewer: 1

Comments to the Author(s)

Finally, please check whether you put all figures and tables information in the proper place in the discussion part? also check whether all references you cite appear simultaneously in main text parts and references part.

Authors response: We checked the five figures and two tables in the manuscript. Only put figure 5 into the discussion part. All the references in the manuscript were checked again.